# Current Trends in Surgical Management of Hepatocellular Carcinoma

**DOI:** 10.3390/cancers15225378

**Published:** 2023-11-12

**Authors:** Isabella Angeli-Pahim, Anastasia Chambers, Sergio Duarte, Ali Zarrinpar

**Affiliations:** Department of Surgery, College of Medicine, University of Florida, Gainesville, FL 32608, USA; isabella.angelipahim@surgery.ufl.edu (I.A.-P.); agchambers@ufl.edu (A.C.); sergio.duarte@surgery.ufl.edu (S.D.)

**Keywords:** hepatocellular carcinoma, surgical management, liver transplant, liver resection, ablation

## Abstract

**Simple Summary:**

Hepatocellular carcinoma (HCC) is the third main cause of cancer-related deaths worldwide, posing a significant global health problem. Surgical management offers the best chance of cure, making it crucial to accurately identify eligible patients and ensure optimal treatment. This paper discusses the ongoing debate regarding criteria expansion and highlights novel techniques for downstaging the tumor and optimizing the future liver remnant, aiming to enable patients to become eligible for surgical therapy. Moreover, the exploration of minimally invasive surgery, improved visualization techniques, and improved hemorrhage prevention techniques showcase promising advancements in HCC surgery that may reduce operative time, surgical stress, and morbidity. This review provides valuable updates for researchers and healthcare professionals, and the findings may shape future research directions, leading to improved outcomes and potentially expanding the pool of eligible patients for curative surgical interventions.

**Abstract:**

Hepatocellular carcinoma (HCC) is a leading cause of cancer-related deaths worldwide. Surgical management, including hepatic resection, liver transplantation, and ablation, offers the greatest potential for a curative approach. This review aims to discuss recent advancements in HCC surgery and identify unresolved issues in the field. Treatment selection relies on the BCLC staging system, with surgical therapies primarily recommended for early-stage disease. Recent studies have shown that patients previously considered unresectable, such as those with portal vein tumor thrombus and uncomplicated portal hypertension, may benefit from hepatic resection. Minimally invasive surgery and improved visualization techniques are also explored, alongside new techniques for optimizing future liver remnant, ex vivo resection, and advancements in hemorrhage control. Liver transplantation criteria, particularly the long-standing Milan criteria, are critically examined. Alternative criteria proposed and tested in specific regions are presented. In the context of organ shortage, bridging therapy plays a critical role in preventing tumor progression and maintaining patients eligible for transplantation. Lastly, we explore emerging ablation modalities, comparing them with the current standard, radiofrequency ablation. In conclusion, this comprehensive review provides insights into recent trends and future prospects in the surgical management of HCC, highlighting areas that require further investigation.

## 1. Introduction

Liver cancer is the sixth most commonly occurring cancer and third highest cause of cancer mortality worldwide, making it a significant global health problem. In the year 2020 alone, over 900,000 new cases and 830,000 deaths were reported worldwide, and its incidence and mortality are expected to increase an additional 50% over the next two decades [1,2]. Liver cancer consists mostly of two major histologic types: hepatocellular carcinoma (HCC) and cholangiocarcinoma (CCA), with HCC making up 80% of cases worldwide [3]. HCC is usually correlated with the presence of an identifiable underlying disease, such as hepatitis B or C, alcohol use, and metabolic-dysfunction-associated fatty liver disease (MAFLD, previously termed non-alcoholic fatty liver disease). Despite advancements in healthcare, the prognosis, even in developed countries, remains poor [1].

Disease stage at patient diagnosis is the primary determinant of treatment options for HCC. Surgical treatment offers the highest rate of cure and survival. However, these are currently reserved for patients with early-stage HCC and minimal background liver disease. In cases where surgical management is not possible, non-curative treatments exist, such as systemic therapies, or locoregional treatments like ablation, transarterial chemo- or radio-embolization, or external beam radiation. These therapies may also be used for downstaging, allowing for later resection or transplantation in previously ineligible patients [1,4]. New developments in surgical management focus on minimally invasive techniques, augmenting surgical and anatomic information through technology, and expanding the boundaries of surgical intervention. The high mortality rates and increasing prevalence of HCC highlight the urgency to explore new therapeutic strategies that may improve patient prognosis. This article aims to review recent developments in the surgical management of HCC and to identify currently unresolved issues. 

## 2. Staging and Treatment Selection

Management of HCC is challenging as it predominantly develops in the context of cirrhosis, a severe underlying disease, necessitating a comprehensive approach to staging and treatment. Considering this, the Barcelona Clinic Liver Cancer (BCLC) staging system has emerged as the preferred approach for staging HCC when compared to the TNM system as it not only considers tumor characteristics but also incorporates liver function and health status into the classification. Therefore, the BCLC staging system provides a more accurate prognosis assessment and is endorsed by American and European guidelines as a mainstay for treatment decisions.

The BCLC staging system stratifies HCC into five categories: BCLC 0, A, B, C, and D. Stages 0 and A encompass patients with early-stage disease, preserved liver function (Child–Pugh A and B), and good performance status. Surgical management is the treatment of choice for patients in this group. BCLC-B represents the intermediate stage and includes patients with multinodular tumors and adequate liver function. Transarterial chemoembolization (TACE) was the only recommended therapeutic approach until the BCLC group published its 2022 update, where it stratified that BCLC-B stage into three groups based on tumor burden and liver function [5]. The first subgroup of BCLC-B is composed of potential candidates for liver transplantation (LT). The second subgroup includes patients not eligible for LT but with well-defined nodules; they are more likely to benefit from TACE. The third subgroup is composed of patients with diffuse, infiltrative, and extensive HCC with bi-lobar hepatic involvement. This group of patients does not benefit from TACE and should receive systemic therapy. BCLC-C corresponds to the advanced stage of HCC, comprising patients with tumors that exhibit portal invasion or extrahepatic spread, along with an impaired performance status. Individuals in this group should be treated with systemic therapy as tolerated. The BCLC-D stage comprises patients with severely impaired liver function who are not eligible for transplantation. BCLC-D is classified as a terminal stage, and these patients should receive best supportive care [2,4,5,6]. Although the guidelines provide a preferred treatment option for each stage, the BCLC 2022 update introduced the concept of treatment stage migration (TSM) that must be taken into consideration when selecting a treatment modality. TSM is applied when the characteristics of a particular patient, such as age, comorbidities, patient values, and availability, influence a change in the recommended approach to the option typically reserved for a more advanced stage [5]. This review focuses on the latest advancements in surgical management of HCC, which specifically targets patients classified as BCLC-0, BCLC-A, and a subset of patients within the BCLC-B stage.

## 3. Downstaging Therapy

### 3.1. Locoregional Therapies

Aside from expanding the criteria, downstaging is currently being implemented to allow previously excluded HCC patients to be treated with surgical management, such as liver resection or transplant. The aim of downstaging is to reduce the tumor burden of higher-stage BCLC patients until they are reclassified as a stage that falls within the surgical management criteria. Non-surgical therapies such as transarterial chemoembolization (TACE), transarterial radioembolization (TARE), and stereotactic body radiotherapy (SBRT) have been used for decades to downstage tumors [7,8]. TACE involves transcatheter delivery of chemotherapeutic drugs into the blood vessel supplying the tumor, followed by embolization agents, resulting in the blockage of the vessel and shrinkage of tumor [7]. This technique is optimal for BCLC-stage B HCC patients with large or multinodular HCC tumors [8]. TARE involves the intraarterial delivery of radioactive Yttrium-90 microspheres to the tumor; however, unlike TACE, TARE induces cell death and tumor shrinkage without disruption of blood flow [9]. Some institutions use TARE in place of TACE for non-ablatable and non-resectable HCC tumors [10]. SBRT is an external beam radiotherapy that uses concentrated radiation doses with extremely precise delivery of radiation to tumors, resulting in minimal damage to surrounding healthy tissues. This technique is generally used for HCC tumors < 5 cm that are not located near critical structures [8]. 

### 3.2. Systemic Therapies

For over a decade, sorafenib was the only clinically effective systemic therapy for treating advanced-stage HCC [11]. However, over the past few years, rapid advancements in molecular targeted therapies, immunotherapies, and combination therapies have revolutionized the treatment of advanced-stage HCC [11]. A series of drugs were developed to inhibit specific molecular abnormalities associated with cancer progression. Some of these clinically approved drugs include levantinib, regorafenib, cabozantinib, and ramucirumab. Lenvatinib, a multikinase inhibitor of platelet-derived growth factor receptor (PDGFR) α, vascular endothelial growth factor receptor (VEGFR) 1–3, fibroblast growth factor receptor (FGFR) 1–4, and proto-oncogenes (RET and KIT), was found to significantly improve overall survival (OS) and progression-free survival (PFS) in addition to providing an increased tumor response, such as objective response rate (ORR) and disease control rate (DCR), as a first-line therapy for unresectable HCC compared to sorafenib [12]. Regorafenib, a multikinase inhibitor of VEGFR 1–3, PDGFR, FGFR, and proto-oncogenes (RAF, KIT, RET, TIE-2), was found to significantly improve OS, PFS, DCR, and ORR compared to placebo in patients previously treated with sorafenib [13,14]. Cabozantinib, a multikinase inhibitor of VEGFR 1-3 and proto-oncogenes (MET and AXL), treatment of previously treated patients with advanced HCC was found to increase OS and PFS compared to placebo [15]. Ramucirumab, a human IgG monoclonal antibody of VEGFR-2, was found to significantly improve OS and PFS compared to placebo for patients who had previously been treated with sorafenib [16].

Occurrence of immune tolerance in HCC patients is due to enrichment of regulatory T cells (Tregs), myeloid-derived suppressor cells (MDSCs), and alteration of immune checkpoint molecules, such as T-lymphocyte-associated protein-4 (CTLA-4) and programmed cell death protein-1 (PD-1) [11,17]. This has led to the development of immunotherapies as a potential therapeutic for advanced HCC. Two immune checkpoint inhibitor monotherapies for HCC treatment have been approved by the FDA, nivolumab and pembrolizumab, which implement an IgG4 monoclonal antibody (mAb) that binds to PD-1, blocking interaction with its ligands, PD-L1 and PD-L2 [18,19]. Nivolumab did not significantly increase OS compared to sorafenib as a first-line treatment; however, a favorable safety profile and clinical activity were observed for patients with advanced HCC [18]. In advanced HCC patients who received previous treatment, pembrolizumab was found to significantly improve OS, PFS, and ORR compared to placebo [19].

In addition to these monotherapies, combination therapies have been developed to achieve a higher therapeutic response. Nivolumab has been used in combination with ipilimumab, recombinant human IgG1κ mAb for CTLA-4, which blocks interations with its ligands (CD80 and CD86), for treatment of advanced HCC patients previously treated with sorafenib [20]. Treatment with nivolumab plus ipilimumab had promising tumor response and had high overall survival rates [21]. For patients with unresectable HCC who have not been treated previously, atezolizumab, an anti-PD-L1 mAb, in combination with bevacizumab, an anti-VEGF mAb, was found to have improved OS and PFS compared to sorafenib [22]. An additional study found that, in patients who had previously been treated, atezolizumab plus bevacizumab had similar beneficial outcomes [23]. Atezolizumab has also been tested in combination with cabozantinib, a multikinase inhibitor of VEGFR, MET, and TAM kinase family (TYROS3, AXL, MER) [24]. Atezolizumab plus cabozantinib significantly improved PFS compared to sorafenib as a first-line therapeutic in patients with advanced HCC [24]. Combination therapy durvalumab, IgG1κ mAb for PD-L1, and tremelimumab, anti-CTLA-4 mAb, as a first-line therapy significantly improved OS compared to sorafenib for patients with unresectable HCC [25]. In patients with intolerance or resistance to sorafenib, durvalumab plus tremelimumab had a promising risk–benefit profile [26]. Cotreatment of camrelizumab, anti-PD-1 mAb, and revoceranib, VEGFR-2 tyrosine kinase inhibitor, for unresectable HCC patients demonstrated increased PFS and OS compared to sorafenib as a first-line therapy [27].

These therapies also hold potential in HCC downstaging, converting previously unresectable cases into resectable ones. In a recent study, patients with unresectable HCC underwent immunotherapy regimens of anti-angiogenic kinase inhibitors (lenvatinib or apatinib) in combination with anti-PD1 antibodies (nivolumab, pembrolizumab, sintilimab, or camrelizumab), and 16% of the patients successfully achieved eligibility and were surgically treated [28]. It is worth noting that this study reported one fatal instance of postoperative liver necrosis, which was attributed to immune-related adverse effects [28]. This highlights the need for the future studies to address liver tissue viability and the safety of the employment of immunotherapy in downstaging, especially in cases where major resection is required. The atezolizumab plus bevacizumab combination is widely used, with a proven positive impact on survival [22,23]; however, its effectiveness in the realm of downstaging is yet to be investigated. While several case studies have shown impressive tumor response and conversion to surgical treatment [29,30], high-impact research is still needed. A prospective phase II trial, named the RACB study, is close to commencing and will investigate the efficacy of this combination in achieving resectability [31]. 

Strategies such as kinase inhibitors and immune checkpoint inhibitors can provide extension of survival and possibly improved resection rates; however, the main problem with these therapies is the development of resistance. This has given rise to a new field of HCC immunotherapy that focuses on promising gene targets and advanced techniques, such as chimeric antigen receptor (CAR)-T cell, dendritic cell vaccine, and oncolytic virus therapy. CAR-T cell therapy modifies the immune system by introducing engineered autologous T cells that recognize and eliminate tumor cells. CAR-T cell therapy is still in early-phase clinical trials for treatment of HCC, but multiple studies have shown promising results to date [32]. Dendritic cells are the most potent antigen-presenting cells in the human immune system and play a role in the recognition and elimination of tumor cells. Consequently, dendritic-cell-based vaccines have been considered in HCC therapy, and, while most vaccines are in early clinical trial phases, this therapeutic holds potential as a personalized medicine [33]. Oncolytic virus therapy uses the replicative ability of viruses to specifically target tumor cells. The viruses selectively infect tumor cells, then replicate within the cell, resulting in immune response causing immunogenic cell death for the tumor cell or even stimulating an antitumor response [34]. Oncolytic virus therapy has a range of clinical trial phases currently recruiting patients. Overall, recent advancements in targeted therapies and immunotherapies have provided a wide range of beneficial effects for previously unresectable advanced HCC patients. Further discussion of downstaging for liver transplantation will be presented subsequently.

## 4. Surgical Management of HCC

Hepatic resection, liver transplantation (LT), and ablation are the surgical therapies considered curative for HCC. They are associated with the most promising outcomes, with 5-year survival rates of 70–90%. The choice of surgical management of HCC requires careful consideration of factors such as tumor size and involvement of major vascular structures, as well as the patient’s liver function and clinical status [1,4].

### 4.1. Hepatic Resection

Evaluating the resectability of a lesion involves evaluation of tumor number and location and hepatic reserve. Surgical resection is the recommended treatment for patients with solitary tumors, preserved liver function, and no evidence of portal hypertension (normal bilirubin, hepatic venous pressure gradient < 10 mmHg, and platelet count > 100,000/µL). In these cases, surgical resection offers low mortality and is associated with a 5-year survival rate of nearly 70%. There is no strict size cutoff for tumor diameter, and even large tumors can be safely resected as long as there is sufficient hepatic reserve. The main complications of resection include hepatic failure and recurrence; therefore, surveillance should be performed every 3 to 6 months by imaging and serum alpha-fetoprotein (AFP) measurements [2,4].

#### 4.1.1. Expansion of Selection Criteria

HCC usually presents without symptoms, leading to delayed diagnosis in most cases. At the time of diagnosis, 70–80% of patients have HCC advanced to the point of unresectability [35]. The main contributing factor for this delay is that the major symptoms of HCC (such as jaundice, ascites, and other signs of portal hypertension) result from macrovascular invasion, most often portal vein tumor thrombus (PVTT), which signifies an advanced disease stage (BCLC-C) [35].

The current guidelines rule patients with PVTT as unresectable; however, with PVTT reaching an incidence of up to 62%, this excludes a large number of patients from surgical resection [35]. For these cases, systemic therapy with molecular-targeted drugs is the recommended treatment [2,4]. Large retrospective studies conducted in Asian populations have demonstrated that surgical treatment can lead to significantly longer overall survival times in patients with preserved liver function as long as the PVTT is limited to a first-order branch of the portal vein [36,37]. In a large retrospective study comparing patients with PVTT treated with liver resection (*n* = 2093) to those who received non-surgical treatment (*n* = 4381), Kokudo et al. found that patients who underwent liver resection survived almost 2 years longer than the group who received other therapies (2.87 vs. 1.10 years, *p* < 0.001) [36]. Asian guidelines already consider surgery to be the preferred treatment in patients with PVTT in first-order branches of the portal vein, preserved liver function, and performance status [35]. These findings emphasize the need for further investigation in Western countries as they may warrant a potential expansion of resection criteria. 

The presence of clinically significant portal hypertension (CSPH) is a contraindication for surgical resection as studies have associated it with increased mortality and postoperative complications [38]. However, recent studies have shown that laparoscopic resection can be safely performed in well-selected patients with CSPH, including patients with a minor degree of CSPH, platelet counts exceeding 100,000/µL, non-decompensated cirrhosis, and favorable tumor conditions for laparoscopic resection, such as small size or peripheral location [5,39]. Further investigations should be conducted as they may warrant the expansion of the current surgical indications to include patients with CSPH who would have previously been deemed unresectable.

#### 4.1.2. Optimizing Future Liver Remnant

Post-hepatectomy liver failure (PHLF) is one of the most severe potential complications of hepatic resection and is considered to be the main cause of morbidity and mortality associated with hepatectomy. Optimizing future liver remnant (FLR) is a strategy that may increase the number of patients who are eligible for surgical resection and improve postoperative outcomes [40,41]. It is believed that the minimal FLR must be approximately 20% of the original hepatic volume in patients with a normal liver, 30% in patients with liver injury (e.g., chemotherapy-associated injury or steatohepatitis), and 40% in patients with cirrhosis [42]. The FLR volume can be estimated by cross-sectional imaging (CT or MRI scans) with three-dimensional reconstruction. Total liver and tumor volumes are measured and used to calculate the FLR volume [43]. In addition to FLR volume, it is also important to estimate FLR function.

A multitude of diagnostic and prognostic techniques have been implemented over the last few decades to more accurately estimate liver function in both acute and chronic liver disease patients. While a clinical examination or biopsy of the patient’s liver is the most common diagnostic tool for evaluation of FLR function, recently, more molecularly driven, non-invasive techniques have been implicated, such as imaging, chemical biomarkers, and enzymatic tests [44]. One such non-invasive imaging technique introduced over the last decade is ultrasound-based elastography (transient or shear-wave), which uses low-frequency vibrations to measure the stiffness of the liver tissue and determine the extent of fibrosis [45,46,47,48]. Since an increase in scarring of the parenchyma is associated with the progression of disease, elastography can not only be used for initial evaluation of disease severity but also serve as a technique to monitor disease progression. Ultrasound is often combined with techniques such as magnetic resonance imaging (MRI) and computed tomography (CT), which assess cirrhosis of the liver and its associated complications [49,50,51]. One of the more common non-invasive techniques uses conventional blood parameters to score patients’ levels of “serum biomarkers” of interest, including bilirubin, albumin, and coagulation proteins, to help evaluate the patient’s stage and severity of disease. However, no singular biomarker can be used to make definitive conclusions; therefore, they must be used in combination, with other clinical parameters, or validated via biopsy in order to properly score the extent of the FLR function [44,49,50,51]. This is traditionally evaluated through the Child–Pugh score, the model for end-stage liver disease (MELD) score, and liver function tests.

Currently, the dominant determinant of FLR function and liver graft utilization is physician evaluation; however, this is variable and difficult to standardize. Therefore, a standardized technique to maximize use of potentially overlooked good grafts and minimize the evaluation of unusable grafts is essential. Tests measuring enzymatic liver function have been implemented as a potential technique to address this issue. Over the past decade, the indocyanine green (ICG) test has gained increasing relevance and is endorsed by current guidelines [4]. ICG is a water-soluble fluorescent dye that hepatocytes selectively uptake then excrete into the bile duct without significant enterohepatic recirculation or extrahepatic elimination [52,53]. Its retention rate at 15 min can be measured at bedside by a non-invasive pulse dye densitometry, and the ICG clearance can indicate the status of hepatic blood flow and hepatocellular function [54,55]. ICG clearance has been used in real time for living and deceased donor liver transplantation to evaluate FLR function intraoperatively and post transplantation to determine graft dysfunction [56,57]. While ICG is not yet sensitive enough to function as a stand-alone assessment, in combination with other methods of evaluation, it can provide improved estimates of liver function in a timely, non-invasive manner.

More recently, liver maximum capacity (LiMAx) breath test has been introduced as a non-invasive technique to determine enzymatic liver function by measuring the hepatic cytochrome P450 1A2 metabolism of a ^13^C-labelled substrate, methacetin [58]. The metabolism of ^13^C-methacetin produces ^13^CO_2_ as a byproduct, which is measured and used to calculate a ^13^CO_2_: ^12^CO_2_ ratio. A lower ^13^CO_2_: ^12^CO_2_ ratio is indicative of a reduced metabolic rate, which corresponds with impaired liver function. The respective volume of functional liver resected during surgery was found to have an equivalent decrease in the LiMAx value [59]. Since LiMAx values were highly correlated to the patients’ functional liver volume, this test can be used to accurately measure liver function capacity [44]. The LiMAx test has successfully provided real-time measurements of changes in patients’ liver function; however, this can be further validated using transient elastography to increase diagnostic accuracy [60].

Recently, the model for end-stage liver disease (MELD) score, which is widely used in liver transplantation, was used in combination with estimated FLR volume to create a score that can predict post-hepatectomy liver failure in HCC [61]. Similarly, many other methods have been recently proposed as useful tools for PHLF risk assessment, such as the combination of Child–Pugh score with FLR measurements [62] or measurement of serum hyaluronic acid levels [63], although these techniques require further validation.

Patients who are determined to be unresectable by the aforementioned tests due to insufficient FLR volume may be eligible to receive treatments that stimulate hypertrophy of the FLR, which may lead to them becoming resectable [43]. Portal vein embolization (PVE) was proposed around three decades ago as a technique to induce FLR augmentation [64]. PVE is considered an effective method that causes a mean increase in FLR volume of 38% [43]. However, due to insufficient FLR augmentation and tumor progression, approximately 15% of patients undergoing PVE remain unresectable [65,66]. The main limitation of PVE is the slow pace of increase in FLR volume of 1.6% per day [67], requiring 4–7 weeks to reach a sufficient increase in volume for resection to be performed, during which tumor progression can occur [43,66,68]. 

Since it was introduced in 2012, Associating Liver Partition and Portal Vein Ligation for Staged Hepatectomy (ALPPS) has gained relevance as an alternative method to obtain FLR hypertrophy. It consists of performing the liver resection in two stages (Figure 1). Ligation of the portal vein and an in situ division of the parenchyma are performed in the first stage. When sufficient FLR has been achieved, the remaining structures are transected and the tumor bearing liver is removed in the second stage [69]. The addition of in situ segmentation is proposed to enhance hypertrophy via two pathways: first, the trauma caused by the transection increases systemic release of factors that induce rapid proliferation (such as IL-6 and TNF-α); second, the addition of in situ segmentation also prevents the formation of vascular collaterals between the two portions of the liver [65,67,70]. This allows for much faster FLR hypertrophy, with studies showing augmentation of nearly 88% [69]. 

A recent study by Chen et al. investigated the effectiveness of ALPPS in HCC patients, with an average growth of 59% after a mean of 16 days, with almost 92% of the patients completing the second stage [71]. In a clinical trial comparing ALPPS to PVE in hepatitis-related HCC, ALPPS induced a 49% increase in FLR volume, and patients required a median of 7.5 days until the second stage could be performed [67]. In this study, ALPPS allowed for a significantly higher rate of resection when compared to PVE (97.8% vs 67.7%, *p* < 0.001) while maintaining comparable mortality and postoperative complication rates (20.7% vs 30.4%, *p* = 0.159) [67]. Meanwhile, a separate study of patients with advanced HCC undergoing open-approach ALPPS revealed a significant complication rate of 69.2% after Stage 2 [72]. 

Despite studies showing variable outcomes, it is undeniable that a major limitation of ALPPS is its associated morbitidy rates [71,72]. This prompted the development of minimally invasive (MI) approaches due to the well-known benefits of laparoscopic or robotic techniques (further discussed in Section 4.1.3). Cioffi et al. [69] performed a systematic review that demonstrated that MI-ALPPS is an effective approach, with a median 87.8% increase in FLR and a median of 21 days in between stages. In this study, the percentage of patients with major complications (defined as Clavien–Dindo Classification of 3a or higher) was 23.4% after Stage 2. However, only 21% of patients that underwent a minimally invasive approach in Stage 1 proceeded with a minimaly invasive Stage 2. This is likely due to the increased technical difficulty of Stage 2, which is associated with higher blood loss and complication rates. The complexity and novelty of ALPPS caused an initial hesitancy to adopt laparoscopic or robotic approaches. Likely as a consequence, this study was limited by the scarcity of literature, including six papers, mostly descriptive case series, with a total of 119 patients [69]. This question is far from resolved as an analysis of data from the ALPPS Italian Registry from 2012 to 2021 revealed that the utilization of MI-ALPPS surged from 3.3% to 46.4% in Stage 1, and from 3.4% to 30.4% in Stage 2 [73]. The study noticed that a minimally invasive Stage 1 was associated with a decreased 90-day mortality. They also found that performing a minimally invasive approach significantly reduced overall morbidity and increased the rate of discharge after Stage 2 (from 15.7% to 82.2%). An additional noteworthy finding was the decrease in post-hepatectomy liver failure from 45.5% to 26.1%, and in major morbidity from 52.4% to 28.5% [73]. While a direct causal relationship between these improvements and the rise in MI-ALPPS cannot yet be confirmed, it offers promising prospects for future developments of this FLR hypertrophy tool. 

#### 4.1.3. Minimally Invasive Surgery

Like many other surgical procedures, minimally invasive resection of HCC is increasingly being adopted. Initially, laparoscopic liver resection (LLR) encountered skepticism, with concerns about the adequacy of resection margins and the potential for less radical resection when compared to open liver resection (OLR). The limited ability to palpate the margins was one of the reasons behind these concerns [74,75]. Surgeons also faced technical challenges, particularly in managing hemorrhage during laparoscopic procedures [76]. As laparoscopic operations have become more common and surgeons have gained experience, these limitations have gradually been overcome. Today, there is extensive literature demonstrating the benefits of LLR over OLR [77,78]. In a prospective clinical trial, El-Gendi et al. randomized 50 patients with solitary HCC < 5 cm to OLR or LLR. The LLR group had significantly shorter operative time (OT) and length of hospital stay while maintaining comparable oncological outcomes. The rate of intraoperative complications was similar between the groups, as were the rates of recurrence and disease-free survival, countering previous concerns [79]. Similar results were found in an international study of patients with HCC and portal vein hypertension, with lower lengths of hospital stay in the group who underwent minimally invasive surgery and comparable oncological outcomes [80].

To address the lack of trials involving major hepatectomies (resection of at least three liver segments) indicated for large tumors (>5 cm) located centrally or near major hepatic vessels, the AP-LAPO (Asia-Pacific multicenter randomized trial of laparoscopic versus open major hepatectomy for HCC) is currently open. The primary outcome of this trial is recurrence-free survival, and the final reports will be published after completion of the 5-year follow-up period, projected to be completed in 2029 [81]. Additionally, laparoscopic HCC resection is associated with significantly reduced pain in the surgical site, as well as lower wound infection rates, when compared to open resection [75,82].

Robotic liver resection (RLR) promises to be a safe and feasible approach to minimally invasive surgery. It implements the use of highly flexible robotic arms, which allows one to overcome movement limitations of laparoscopic surgery; it additionally provides benefits such as enhanced precision and minimal abdominal wall straining. Compared to LLR, RLR is associated with higher cost, longer operative times, and higher transfusion requirements; however, RLR has a lower rate of conversion to open procedure [83]. Currently, RLR is not performed in many centers but is gaining popularity as it becomes more accessible. The safety of minimally invasive hepatic resection, laparoscopic [84] or robotic [85], in patients with prior abdominal operations was recently established, addressing previous concerns about these procedures in the presence of adhesions. Overall, minimally invasive procedures reduce surgical stress, improve patient recovery, and should be considered whenever feasible [5].

#### 4.1.4. Visualization Techniques

With the increasing adoption of laparoscopic and robotic procedures, there has been a corresponding emergence and improvement in technologies aimed at enhancing visualization. Among these advancements, the combination of augmented reality navigation (ARN) and fluorescence imaging has proven to be a valuable tool set in assisting hepatectomy procedures. These techniques may improve outcomes and help to decrease the rates of conversion to open operations.

As discussed previously, ICG is a fluorescent dye that can be used in the assessment of FLR. Nevertheless, it has other important applications in the context of intraoperative management of HCC. The use of fluorescence imaging after intravenous administration of ICG allows for intraoperative visualization of liver tumors (Figure 2) [86]; however, its sensitivity is influenced by tumor and technical factors. A study showed that ICG-fluorescence imaging identified 51% of HCC sites intraoperatively. However, the method had 100% sensitivity when applied on the resected specimens. They found that the tumors that were not identified intraoperatively through the technique were significantly smaller (11 mm vs. 18 mm, *p* = 0.019) and located deeper in the liver (10 mm vs 2 mm, *p* < 0.001), not identifying any of the tumors that were deeper than 8 mm from the liver surface [87]. This method can also be used to identify the bile duct [88], a structure that, if injured, causes intra- and postoperative complications, such as bile leak and biliary stricture (Figure 3) [89]. A recent study has shown that decreasing the ICG dose to 0.05 mg, instead of the standard 2.5 mg, preoperatively decreases hepatic and background fluorescence, allowing for a clear bile duct visualization within 15–20 min from administration [88].

For over a decade, three-dimensional visualization has been used in pre-operative assessment for resection, where a “virtual hepatectomy” is performed to allow for measurement of liver segmental volumes and estimation of FLR. This technique uses CT images to create a 3D reconstruction of the anatomy and can help to determine a tumor’s resectability, as well as a surgical plan [90]. Recently, studies have explored the possibility of using 3D models in real time during surgery, an approach named augmented reality navigation (ARN) [91,92].

ARN systems facilitate identification of the anatomy and spatial distribution of the liver, vascular, and biliary structures due to the depth perception provided by 3D visualization, as well as the assignment of colors for each type of structure [92]. Zhang et al. recently published their preliminary experience with the use of ARN and found that the resections where they used the visualization system had significantly lower intraoperative blood loss, decrease in hemoglobin, and blood transfusion rates when compared to the resections where they did not use the technology. The ARN group also had accelerated postoperative recovery with significantly shorter postoperative lengths of hospital stay [92].

By facilitating identification of important structures, these technologies have the potential of overcoming limitations of resecting tumors located in challenging areas [91]. Zhu et al. conducted a study to assess the efficacy of the combination of ARN and ICG-fluorescent imaging for laparoscopic resection of centrally located HCCs. These are the lesions situated in the left medial and right anterior sections, which are more prone to hemorrhage and biliary leakage due to their location adjacent to important structures. The group found that the patients undergoing surgery with the combination of ARN and fluorescent imaging had significantly reduced intraoperative blood loss and transfusion rates, shorter postoperative length of stay, and lower perioperative complication rates when compared to patients undergoing routine laparoscopic hepatectomy. They also found that the combination of methods significantly reduced the rate of conversion to laparotomy (35.7% vs. 61.8%, *p* = 0.024) [91].

#### 4.1.5. Ex Vivo Resection

Patients with lesions infiltrating large vascular structures, or lesions located in difficult anatomical sites, are often ineligible for conventional resection [93]. A technique known as ex vivo liver resection and autotransplantation (ERAT) is implemented for these patients with conventionally unresectable liver tumors [94,95]. ERAT was first performed by Pichlmayr et al. in 1988 on a patient with bilateral liver metastases, which were previously regarded as unresectable, with good liver function after reimplantation [96]. The procedure involves a total hepatectomy, vascular reconstruction, and surgical resection on the externally cooled liver on a surgical table, and an autologous transplantation of the liver remnant [93,97]. Due to the autotransplant aspect of the procedure, ERAT is traditionally reserved for non-cirrhotic patients with good liver function [98]. The extracorporeal aspect of ERAT allows for a bloodless field to perform the resection of the tumor(s) in a decreased time pressure setting [93]. Since the initial surgery, ERAT has been used in selective cases for patients with various types of liver tumors [93], and a systematic review conducted by Weiner et al. found overall survival to be 67%, 39%, and 28% for patients at years one, three, and five, respectively [97]. Additionally, patients with tumors categorized as low-grade had a 100% survival rate at five years post ERAT [97]. While the surgical risk of ERAT is significant, the overall survival in the surgical cohort was far superior to that of the non-surgical cohort, 82.1% versus 19.1% five years post-surgery [99]. Over the last few decades, surgical progress and experience have allowed ERAT to become an established procedure; however, an elevated risk of perioperative failure and mortality still remains [100]. In the case of surgical complication or remnant liver failure, a patient may require liver allotransplantation [93]. Therefore, ERAT is a complex, radical procedure reserved for a select group of patients with normal liver function and conventionally unresectable hepatic lesions.

#### 4.1.6. Hemorrhage Prevention and Control

Intraoperative blood loss is directly associated with a higher risk of postoperative complications; employing different surgical techniques that focus on minimizing blood loss is a major aspect of hepatic resections. Currently, the intermittent Pringle maneuver (IPM) is the most common and well-established technique performed in hepatic operations. It aims to reduce blood loss during liver resection and involves repeated cycles of temporary clamping of the hepatic inflow for 15 to 20 min, followed by a 5-minute perfusion. In recent years, important strategies for controlling blood loss have emerged, including maintaining a low central venous pressure, which is often used in combination with IPM [101].

A recently published systematic review of 18 randomized control trials (RCTs) including 1285 subjects who underwent hepatectomy concluded that maintaining a controlled low central venous pressure (CLCVP) under 5 mmHg significantly reduces intraoperative blood loss and the need for transfusion [102]. CLCVP can be achieved using various strategies, including fluid restriction or administration of drugs, such as diuretics or nitroglycerin. These methods may cause complications such as hemodynamic instability and electrolyte disturbances. An RCT conducted on 52 subjects proposed milrinone, a medication with vasodilative effects, as a more favorable alternative to achieve CLCVP [103]. Patients receiving milrinone had significantly improved outcomes compared to those receiving nitroglycerin, including reduced blood loss, shorter duration of hepatic hilum occlusion and liver resection, improved intraoperative hemodynamic parameters, and shorter hospital stays [103]. CLCVP is a safe and effective strategy in the prevention of blood loss; however, there is still no consensus regarding the preferred technique to achieve it.

The combination of IPM and CLCVP was used in Li et al., an RCT that aimed to determine the clinical impact of a strategy that would shorten operative time. The study demonstrated in patients with HBV-related HCC that prolonging the hepatic hilum clamping time from 15 min to 20 min significantly reduced intraoperative time while not inducing hepatic injury. Patients in the 20 min group did not show significant differences in postoperative aminotransferase levels or bleeding, showing that it is feasible and safe to prolong the hepatic hilum occlusion time to 20 min using the IPM combined with CLCVP [101]. Implementing this strategy may yield important advantages to the patients as longer operative times are associated with a higher incidence of complications [104].

Recent studies have focused on testing the feasibility and safety of blood loss control surgical techniques that may already be well-established in open surgery in the setting of minimally invasive resection. For example, IPM was recently demonstrated to be a safe technique in minimally invasive hepatic resection for patients with HCC, including those with compensated cirrhosis, showing no differences in postoperative complications or liver function tests [105]. During IPM, the surgeon clamps vascular inflow at the hilum, resulting in total hepatic inflow occlusion (TIO). An alternative technique is hemi-hepatic inflow occlusion (HIO), which selectively interrupts the arterial and venous inflow to a specific side of the liver, aiming to prevent splanchnic blood stasis, global hepatic ischemia, or ischemia-reperfusion injury [106]. Although TIO and HIO are well-established surgical techniques, they are both more challenging to perform in laparoscopic liver resections (LLR) when compared to open liver resections (OLR). The optimal approach for controlling blood loss in LLR has been a subject of debate. However, an RCT was recently conducted by Peng et al. to compare the two techniques in LLR. The study found that there were no significant differences overall between the two approaches, although TIO was considered to be simpler to perform and resulted in a trend towards shorter operative times. Specifically, in cases when the transection plane was on Cantlie’s line, TIO resulted in shorter perioperative time and less blood loss [76].

### 4.2. Liver Transplantation

Liver transplantation is a very effective therapy for HCC as it radically addresses both the tumor and the underlying hepatic disease. It is recommended for patients with advanced cirrhosis, clinically significant portal hypertension, hepatic decompensation, and early-stage HCC within the Milan criteria. It is well established that, for patients with HCC in stages BCLC-0 and A who meet criteria for LT listing, LT should be considered as a first option [2,4]. However, LT has gained a new indication in the BCLC 2022 update; patients with intermediate-stage (BCLC-B) with well-defined HCC nodules should be considered for LT if they meet the institution’s ‘Extended Liver Transplant criteria’ (usually based on size or AFP levels) [5].

#### 4.2.1. Inclusion Criteria

The Milan criteria hold that LT can be performed in patients with a single tumor measuring ≤ 5 cm, or 2–3 tumors that are ≤3 cm each, without major vascular invasion or extrahepatic tumor spread. For decades, this has been considered the gold standard method for HCC patient selection for LT [2,4]. However, with the increasing incidence of HCC and because most patients have exceeded the Milan criteria at the time of diagnosis, this method has been repeatedly challenged as it is perceived as excessively restrictive. Several centers have proposed alternative criteria that may encompass a larger number of patients. Xu et al. reviewed 6012 patients from the China Liver Transplant Registry [107]. They aimed to identify an approach that would expand the selection beyond the Milan criteria while maintaining comparable outcomes. The study demonstrated that the Milan criteria excluded 56% of LT candidates, and that four more inclusive criteria (University of California, San Francisco, University Clinic of Navarra, Valencia, and Hangzhou criteria) provided an expansion in the number of eligible patients ranging from 12.4% to 51.5% while maintaining comparable tumor-free survival and overall survival. An important alternative method of extended liver transplant criteria was the one adopted by Region 4 of the United Network for Organ Sharing (UNOS), where they expanded the criteria to include a single tumor ≤ 6 cm or up to three tumors with the largest diameter ≤ 5 cm and total sum of diameters ≤ 9 cm. After 10 years of implementation of this method, analysis showed that there was no significant difference in 10-year patient survival, recurrence-free survival, or allograft survival when compared to the Milan criteria group. In the meantime, the expansion of criteria allowed for an increase of 9% in patients who were now eligible for LT [108]. While none of the expanded criteria were ever implemented nationwide, the promising outcomes observed in these studies may warrant reconsideration regarding future guideline revisions.

There has also been a growing movement to incorporate markers beyond tumor size alone into the decision of patient eligibility for LT, particularly the use of AFP levels. AFP > 1000 ng/mL is currently the most widely used cut-off value for exclusion from LT eligibility [5]. Mehta et al. [109] demonstrated that a reduction in AFP from >1000 to <500 ng/mL through locoregional therapy leads to significantly reduced HCC recurrence and prolonged 5-year post-LT survival. This study showed that the probability of 5-year recurrence was 35.0% for patients with AFP levels >1000 ng/mL at the time of LT. In comparison, the recurrence rate was 13.3% for individuals whose AFP decreased to the range of 101–499 ng/mL (*p* < 0.001) and 7.2% for those with an AFP decrease to ≤100 ng/mL (*p* < 0.001). Patients who had AFP > 1000 ng/mL at the time of LT had 49% 5-year post-LT survival, compared to 67% for those with a decrease in their AFP to 101–500 ng/mL and 88% for those with decrease in their AFP to ≤100 ng/mL [109]. These findings show that AFP is a valuable marker for prediction of recurrence and survival in HCC patients undergoing LT and would be useful to be incorporated into the treatment selection process. Although consensus on the optimal criteria for LT patient selection in HCC is yet to be reached, the likely direction is that the tumor burden limit will be expanded and that a composite criterion will be implemented, where markers beyond tumor size, such as AFP, will be considered.

#### 4.2.2. Bridging and Downstaging Therapy

One of the main limitations of LT is the shortage of available allografts [2,4,110]. Thus, the wait time for an allograft may be long, and, unfortunately, many HCC patients end up progressing to a more advanced stage, becoming ineligible for transplantation. In an attempt to prevent patients from progressing and reduce the risk of waitlist dropout, bridging therapy is performed. Currently, its indication is justified in LT candidates for whom the waiting time is expected to exceed 6 months [110]. Bridging therapy may be performed via percutaneous ablation, transarterial chemo- or radioembolization, radiotherapy, or a combination or modalities [2]. The technique selected for bridging therapy is dependent on patient condition, such as tumor size, number of nodules, and AFP concentration [111]. To be considered for thermal ablation, such as radiofrequency or microwave ablation, patients must have adequate liver function, no encephalopathy, and ≤3 tumors that are ≤3 cm in size. TACE also requires adequate liver function and no encephalopathy, but tumors ≤15 cm are eligible for use of TACE as a bridging therapy [112]. SBRT is generally used for patients ineligible for TACE or RFA as it can be performed for patients with liver dysfunction. Sapisochin et al. compared outcomes between TACE, RFA, and stereotactic body radiotherapy (SBRT) and found no differences in waitlist dropout rates, postoperative complications, and survival. However, patients in the RFA group presented superior tumor necrosis in the explant and lower risk of recurrence [112]. While current guidelines do not recommend one treatment over any other, the bridging therapy selected is highly dependent on the medical condition of the patient and the number and size of tumors [2]. 

The locoregional therapies used in bridging therapies, naturally, can lead to downstaging with partial or complete response, and there is a current debate on whether patients with tumor response should still undergo LT. The XXL trial [111] demostrated that LT post successful response to downstaging therapy of HCC resulted in increased tumor event-free and overall survival compared to other non-transplantation therapies. Vitale et al. [113] conducted a follow-up study that challenged the XXL trial protocol of prioritizing patients with sustained complete response (i.e., full response not followed by recurrence) to therapy. In this study, they did not provide any priority to these patients and found that they had similar survival compared to the patients who were prioritized in the XXL trial, and also had similar survival to patients who were not treated with LT [113]. This suggests that a “wait and see” approach to patients with complete response could be adequate. This policy has the potential of not only avoiding unnecessary surgical procedures for patients with good prognoses but also potentially sparing donor organs for patients with a more time-sensitive need for LT. 

### 4.3. Ablation

Ablation is a procedure that has rapidly grown in use during the last decade and is regarded as potentially curative for HCC. It aims to achieve destruction of tumor cells by the injection of chemical substances (ethanol, acetic acid, and boiling saline), by modifying local tumor temperature (radiofrequency, microwave, laser, and cryotherapy), or by the utilization of electrical currents. The procedure can be non-surgical (percutaneous using imaging guidance) or surgical (laparoscopic or open). Ablation is currently considered the preferred option for patients with BCLC stage 0 or A who are not candidates for LT [2,4,5].

Takayama et al. recently published the results of the Surgery versus Radiofrequency Ablation for Small Hepatocellular Carcinoma (SURF) randomized controlled trial. The study, conducted in 301 patients from 49 different institutions with the largest HCC diameter ≤3 cm and ≤3 HCC nodules, compared the outcomes of RFA and surgical resection with a median follow-up period of five years. The trial did not find significant differences in recurrence-free survival between the RFA and surgery groups (median of 3 vs. 3.5 years, respectively) and observed advantages in the RFA group, such as shorter procedure duration (40 vs. 274 min, *p* < 0.01) and shorter hospital length of stay (10 vs. 17 days, *p* < 0.01). This supports the recommendation of ablation over resection in cases of very small tumors. However, local recurrence rate tended to be higher in patients in the RFA group, although not statistically significant (28% vs. 15%, *p* = 0.07) [5,114]. Other limitations of RFA include tumor location as it is contraindicated in areas near the main biliary tree, abdominal organs, or heart due to risk of organ damage [2]. 

Radiofrequency ablation (RFA) is currently the most widely used technique as it has long been proven to be superior to percutaneous ethanol injections [115]. RFA involves the insertion of a probe into the tumor site to deliver a high-frequency electrical current that generates heat and results in cell death. Microwave ablation (MWA) is an alternative to RFA. MWA can achieve higher temperatures in shorter periods of time via the use of electromagnetic waves while maintaining comparable oncological outcomes [2,116]. Yu et al. published a clinical trial that compared RFA against MWA in patients with HCC < 5 cm and preserved liver function and showed that MWA required significantly less time (9 vs. 24.4 min, *p* < 0.001), smaller puncture size (2.6 vs. 3.2 cm, *p* < 0.001), and lower hospitalization costs (43,200 vs. 50,300 RMB, *p* < 0.001—approximately 6140 vs. 7150 USD) when compared to RFA [116].

Another ablation technique that has emerged in the past decade is irreversible electroporation (IRE), also known as “NanoKnife”. IRE utilizes high-voltage electrical pulses between two probe pairs to achieve tumor cell death. As a non-thermal form of ablation, it may overcome the limitations associated with tumor location by reducing the risk of adjacent organ or vascular damage and avoiding the heat sink effect that limits the ablative effect of RFA or MWA on a lesion near a large blood vessel. A recent RCT compared IRE with RFA and found comparable outcomes, with similar rates of success, recurrence, and adverse effects in patients with solid hepatic tumors, although IRE was found to be a significantly longer procedure (34 min vs 20 min, *p* < 0.001). These findings suggest that IRE can be considered a safe alternative to thermal ablation [70,117].

## 5. Future Directions and Conclusions

HCC poses a significant global health challenge, demanding innovative approaches to improve patient outcomes. Our investigation of the evolving landscape of HCC surgical management reveals a promising trajectory towards reduced morbidity and imporved survival.

Surgical management of HCC is currently undergoing a transformative shift towards minimally invasive techniques after facing skepticism from surgeons for quite some time. The increased adoption of minimally invasive resections, especially robotic resections with their reduced abdominal wall straining, can significantly reduce patient pain and postoperative recovery time. As robotic procedures become more accessible, we expect surgeons to become more facile with the technique, overcoming the current most important drawback, which is the longer operative time. The integration of cutting-edge technologies additionally simplifies minimally invasive surgical management. Real-time anatomical visualization during surgery, facilitated by ARN and 3D models generated from CT scans, is already a reality. As artificial intelligence continues to advance, we can expect the utilization of enhanced visualization methods and surgical planning, benefitting both surgeons and patients by reducing operative time and enhancing anatomical identification, which can be particularly helpful in cases with tumors in complex locations, or with complex anatomical variations.

Although the surgical management of HCC, including liver resection, transplantation, and ablation, offers the most promising outcomes, the conventional criteria have historically been strictly reserved for patients with preserved liver function and early-stage HCC. Obviously, a meticulous review of patient eligibility for such a major procedure is essential, but the strictness of such criteria has been challenged for quite some time. Although multiple studies have been published showing that select expanded eligibility criteria are safe and can improve mortality, the strict criteria are still the most widely used. With some Asian countries successfully implementing less restrictive criteria, such as allowing resection for a select group of patients with PVTT, we expect a move towards broadening the eligible patient population, likely improving chances of survival. In addition to criteria expansion, novel approaches are evolving to enhance patient eligibility. For example, insufficient future liver remnant is a major exclusion criterion from surgical resection; however, the emergence of ALPPS may change this scenario with its exceptionally accelerated capacity to induce FLR hypertrophy. Likely in the upcoming years, adaptations of ALPPS will aim to reduce morbidity, illustrated by the aforementioned preliminary studies investigating a form of minimally invasive ALPPS. In addition to strategies to increase FLR, advancements in downstaging therapies, especially systemic therapies with the emergence of immune and targeted therapy, have resulted in reduced tumor burden, allowing patients characterized as previously unresectable to undergo surgical management. Specifically in the context of LT, bridging therapy can benefit transplant candidates by attempting to avoid tumor progression beyond eligibility criteria while on a waitlist for an allograft. 

In regards to ablation, percutaneous ethanol injections are on the path of being replaced by RFA and MWA ablation, which have shown superiority in recent years. Additionally, IRE, a novel non-thermal form of ablation, has overcome the concern of tumor-adjacent organ damage. This advancement has the potential of reducing limitations in ablation associated with tumor location. 

In summary, this review provides an overview of the most recent developments in the surgical management of HCC, provides insights about current trends and future possibilities, and emphasizes areas that necessitate additional research. Table 1. provides an overview of the topics discussed in this review along with respective noteworthy publications.

## Figures and Tables

**Figure 1 cancers-15-05378-f001:**
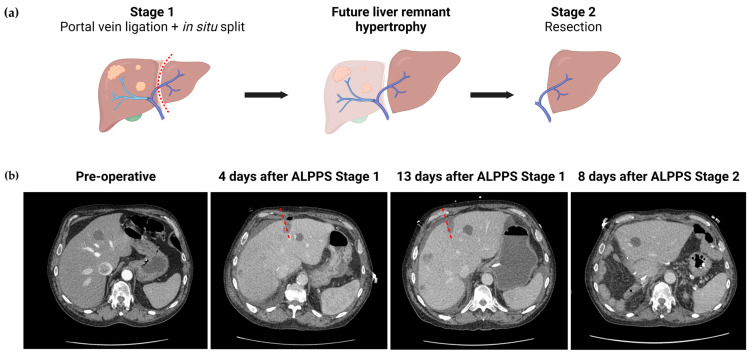
Associating Liver Partition and Portal Vein Ligation (ALPPS) procedure. (**a**) Diagram of ALPPS procedure with right portal vein ligation; (**b**) CT imaging from a patient who underwent ALPPS procedure. In Stage 1, 30–40% of the liver parenchyma was dissected and divided along with ligation of the right portal vein. The dashed line indicates the location of the in situ partition. There was significant left lobe hypertrophy. In Stage 2, extended right lobe hepatectomy was performed.

**Figure 2 cancers-15-05378-f002:**
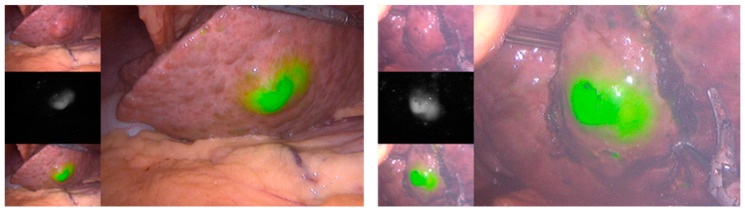
ICG-fluorescent images obtained from two laparoscopic surgeries. The technique highlights the locations of the tumors as fluorescent images.

**Figure 3 cancers-15-05378-f003:**
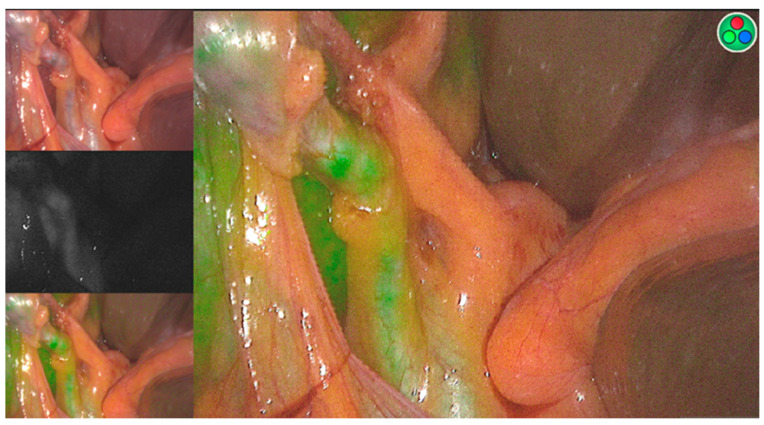
ICG-fluorescent image identifying the common bile duct.

**Table 1 cancers-15-05378-t001:** Overview of Noteworthy Papers and Findings Compiled in this Review.

Reference	Year	Study Population	Findings
Section 4.1 *Resection:* Section 4.1.1 Expansion of selection criteria			
Kokudo et al. [36]	2016	6474 (LR *n* = 2093 and non-LR *n* = 4381)	Patients with PVTT confined to a first-order branch had improved survival with surgical intervention.
Molina et al. [39]	2018	45 (CSPH *n* = 15 and w/o CSPH *n* = 30)	Laparoscopic resection could be safely performed in well-selected patients with CSPH.
Section 4.1 *Resection*: Section 4.1.2 Optimizing future liver remnant			
Buechter et al. [60]	2019	102	The LiMAx test is a non-invasive tool capable of providing real-time measurements of liver function with higher accuracy than TE and serum biomarker.
Chan et al. [67]	2021	148 (ALPPS *n* = 46 and PVE *n* = 102)	ALPPS allowed more patients to undergo resection, promoting faster and greater FLR hypertrophy, while maintaining comparable postoperative and oncological outcomes.
Cioffi et al. [69]	2023	119	Minimally invasive ALPPS demonstrated FLR augmentation of nearly 88%, suggesting to be a feasible and effective approach with the potential of improving the morbidity related to this procedure.
Serenari et al. [73]	2023	268	From 2012 to 2021, the minimally invasive approach to ALPPS increased by 43% in Stage 1, and 27% in Stage 2. This approach was associated with reduced morbidity.
Section 4.1 *Resection:* Section 4.1.3 Minimally invasive surgery			
El-Genti et al. [79]	2018	50 (OLR *n* = 25 and LLR *n* = 25)	LLR provides shorter operative time and length of hospital stay with similar complication rate and oncological outcomes.
Wang et al. [82]	2023	4380 (LLR *n* = 1108 and OLR *n* = 3289)	LLR reduces wound infection, wound pain, and bile leakage when compared to OLR.
Zhang et al. [83]	2020	3544 (RLR *n* = 1312 and LLR *n* = 2232)	RLR was associated with decreased conversion rate, but increased total cost, operative time, and transfusion rate when compared to LLR.
Section 4.1 *Resection:* Section 4.1.4 Visalization techniques			
Zhu et al. [91]	2023	76 (ARN-FI *n* = 42 and non-ARN-FI *n* = 34)	The use of a combination of ARN and fluorescence imaging (FI) in laparoscopic resections was associated with reduced rates of blood loss, conversion to laparotomy, postoperative complications, and hospital stay.
Section 4.1 *Resection:* Section 4.1.5 Ex vivo resection			
Weiner et al. [97]	2022	35	Favorable outcomes for overall survival at 1, 3, and 5 years for patients with low-grade to highly aggressive malignancies suggest that the more liberal use of this technique could benefit selected patients.
Section 4.1 *Resection:* Section 4.1.6 Hemorrhage prevention and control			
Li et al. [101]	2023	151 (Group 20 *n* = 75 and Group 15 *n* = 76)	In resections using IPM combined with CLCVP, extending the hepatic hilum occlusion time from 15 to 20 min resulted in significantly shorter operative times with similar bleeding and postoperative aminotransferase levels.
Section 4.2 *Liver transplantation:* Section 4.2.1 Inclusion criteria			
Xu et al. [107]	2016	6012	The Milan criteria excluded 56% of LT candidates. Meanwhile, 4 alternatives, more included crtieria, expanded the pool of eligible patients from 12.4 to 51.5%, and maintained similar outcomes.
Commander et al. [108]	2018	2068 (Region 4 Criteria *n* = 180 and Milan Criteria *n* = 1888)	10 years after the implementation of expanded LT criteria for HCC patients within UNOS Region 4, there was a 9% rise in LT-eligible patients, with no difference in overall, recurrence-free, or allograft survivals when compared to the Milan criteria group.
Section 4.3 *Ablation*			
Takayama et al. [114]	2022	308 (RFA *n* = 151 and Surgery *n* = 150)	Among patients with small (≤3 cm) and few (≤3) tumors, RFA showed comparable recurrence-free survival, but higher local recurrence rate. RFA was associated with shorter operative time and hospital stay.
Yu et al. [116]	2017	454 (RFA *n* = 251 and MWA *n* = 203)	MWA was associated with lower hospitalization costs, shorter duration, with similar complication and tumor progression rates when compared to RFA.
Zhang et al. [70,117]	2022	152 (RFA *n* = 74 and IRE *n* = 78)	IRE procedures were significantly longer, but demonstrated comparable rates of success, recurrence, and adverse effects to RFA.

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
