# Peer review of "Current Trends in Surgical Management of Hepatocellular Carcinoma"

_cancers, 2023, doi:10.3390/cancers15225378_

Round 1
Reviewer 1 Report (Previous Reviewer 4)
Comments and Suggestions for Authors
The authors have further improved their manuscript. Nonetheless it remains a simple narrative review not bringning anything new in the literature.
Author Response
Thank you for your feedback. We acknowledge the reviewer’s concern. However, we would like to address that this paper was specifically requested as a review paper with a focused topic: “Current Trends in Surgical Management of Hepatocellular Carcinoma”; therefore, our primary aim was to provide a comprehensive and up-to-date overview of the existing literature in this field. We firmly believe that we have successfully created a resource that effectively summarizes the current situation of the current state of the surgical management of HCC, making it a valuable asset for anyone working in the field of cancer treatment. Additionally, we have identified current research gaps, which we believe can contribute to future researchers.
Reviewer 2 Report (New Reviewer)
Comments and Suggestions for Authors
I checked the manuscript.
I understand the reviewer concerned the cited reference. The authors revised the suitable references.
The reviewer also concerned the weakness of this review.
This is very hard to revise completely.
With best,
Author Response
Thank you for your feedback. We are pleased to see that you have given our paper a high score in all categories, and we also appreciate your acknowledgement of the revisions we made to address previous concerns about the cited references.
We have made efforts to incorporate the most recent resources available in the field of surgical management of HCC. Our goal was to provide a strong review paper that aligns with the objectives set by the guest editor who invited us to create this review paper with this specific topic. We agree that the revisions and enhancements we have made contribute to the strength and quality of the manuscript.
Reviewer 3 Report (New Reviewer)
Comments and Suggestions for Authors
Dear editor
the Pahim et al., paper reviews available surgical management protocols regarding the interventions available for hepatocellular carcinoma focusing on inclusion criteria of a wider subgrouping of patients.
The review is well written, although it would significantly improve with more referencing on novel biomarker and immune relate therapies, as this would make it more interesting for the journal as it is very clinically themed.
The format of the review is adequate, the table will need refining as it is difficult to follow in the current from. It looks that the submitted paper still has track changes in the text.
I would change figure 1 with an improved quality as the images are of poor quality and not that informative.
Comments on the Quality of English LanguageThe quality of the English is very good.
Author Response
Thank you for taking the time to review our manuscript and for the constructive feedback. Although we understand the clinical theme of the journal, the invitation from this Special Issue’s editor specifically recommended that we wrote a paper with the topic: “Current Trends in Surgical Management of Hepatocellular Carcinoma”. Because this is a vast and intricate field in itself, initially we purposefully maintained a specific focus on the surgical aspects of HCC treatment. To address this concern, we have thoroughly reviewed the literature and incorporated the most recent papers on new systemic therapies for HCC as they pertain to surgical treatment. We added a section under “downstaging therapies” that is called “systemic therapies”.
Regarding the table, we have made improvements to its format by adjusting line spacing, which we hope addresses your concerns. We have adhered to the table template provided by Cancers. Additionally, we have taken your feedback into consideration and substantially improved the quality of the images in Figure 1. We believe that the diagram and the scans are a great visual representation of the ALPPS procedure, helping readers understand this complex surgical technique.
Reviewer 4 Report (New Reviewer)
Comments and Suggestions for Authors
This review summarizes the current trends in the surgical management of primary liver cancer, HCC. Currently, many studies have shown that other immunotherapy or targeted therapy before surgical treatment provides promising efficacy for advanced HCC treatment, which should be described in the future direction.
Other minors:
Suggest changing “metabolic dysfunction-associated steatotic liver disease” to “Metabolic dysfunction-associated fatty Liver disease (MAFLD)”.
The abbreviation of “hepatocellular carcinoma (HCC)” is shown for the second time in line 63.
T-regulatory cells (Tregs) > regulatory T cells (Tregs).
Author Response
We appreciate your constructive feedback on our review paper, as well as the high scores provided. In response to your suggestion, we have incorporated recent and relevant information about the use of immunotherapy and targeted therapy before surgical treatment of HCC. We have included a new subsection called “systemic therapies”. We have also made the additional edits as suggested. Thank you again for your feedback; we trust that these revisions have strengthened the quality of our manuscript.
Round 2
Reviewer 1 Report (Previous Reviewer 4)
Comments and Suggestions for Authors
The authors yet again present another revised version of their work. It is interesting how the authors feel that because of the fact that they were invited to submit a narrative review article on state-of-the-art management of HCC, their work automatically deserves publication in a journal with an IF of 5.2.
Even if we were to ignore this questionable ?request the authors should think again on what they quote "Additionally, we have identified current research gaps, which we believe can contribute to future researchers." For instance it is interesting how the authors in the bridging treatment to LT paragraph fail once again to cite important studies in the field such as the XXL trial which demonstrated clearly the important role of bridging treatment for LT candidates with HCC. It is unclear to me how "There is not a wide amount of evidence on the efficacy of bridging therapy".
Reviewer 3 Report (New Reviewer)
Comments and Suggestions for Authors
The authors have significantly amended the manuscript hence improving its quality. I recommend it for acceptance.
This manuscript is a resubmission of an earlier submission. The following is a list of the peer review reports and author responses from that submission.
Round 1
Reviewer 1 Report
Comments and Suggestions for Authors
This a well written article on HCC management
However, the glaring absence of TARE for downstaging is a concern
A more detailed description of TACE may also be helpful
Author Response
This is a well written article on HCC management.
- Thank you for your positive feedback on our article.
Point 1: The glaring absence of TARE for downstaging is a concern.
- We incorporated a section on non-surgical therapies, including a discussion on TARE for downstaging. While our primary focus remains on advancements in systematic therapies, we've provided concise descriptions of each non-surgical therapy and its appropriate application.
Point 2: A more detailed description of TACE may also be helpful.
- We elaborated on TACE and its optimal application conditions in lines 104-108. Further discussion on TACE can also be found in the bridging therapies section (lines 518-519).
Reviewer 2 Report
Comments and Suggestions for Authors
I have reviewed the review article entitled "Current Trends in Surgical Management of Hepatocellular 2 Carcinoma". The article is well written and comprehensive. My only comment is that adding a table summarizing the main articles mentioned in this review will add to the manuscript.
Comments on the Quality of English LanguageThe article is well written. Some language polishing will be great.
Author Response
The article is well written and comprehensive.
- Thank you for your positive feedback on our article.
Point 1: Adding a table summarizing the main articles mentioned in this review will add to the manuscript.
- A table summarizing the main articles from each surgical management subsection has been included. Kindly refer to the table below line 604.
Point 2: The article is well written. Some language polishing will be great.
- We have meticulously re-reviewed the manuscript to enhance the language and clarity.
Reviewer 3 Report
Comments and Suggestions for Authors
Overall, this is a thorough and informative summary of surgical management of HCC. Minor suggestions for edits:
-Regarding organization, would consider restructuring articles to what is considered "surgical" therapy for HCC first, and then detail the management by stage. For example, ablation could be considered surgical or non-surgical depending on the operator.
-With respect to content, radiation segmentectomy data with newer dosimetry thresholds have high CPN rates, and ablation is only effective up to a certain size threshold. For example, lines 505-507 should mention the average size of lesions treated in that article.
Author Response
Overall, this is a thorough and informative summary of surgical management of HCC. Minor suggestions for edits:
- Thank you for your positive feedback on our article.
Point 1: Regarding organization, would consider restructuring articles to what is considered "surgical" therapy for HCC first, and then detail the management by stage. For example, ablation could be considered surgical or non-surgical depending on the operator.
- Thank you for the suggestion. We have rewritten the review with the goal of focusing in on the surgical therapies, the brief background, staging and treatment, and downstaging sections are written to set the stage for the surgical techniques, and reorganizing the structure would require extensive rewriting. Additional writing was added to note that ablation can be a surgical or non-surgical technique (line 533-534).
Point 2: With respect to content, radiation segmentectomy data with newer dosimetry thresholds have high CPN rates, and ablation is only effective up to a certain size threshold. For example, lines 505-507 should mention the average size of lesions treated in that article.
- Regarding the bridging therapies compared, the criteria for usage and sizes of lesions each technique is optimal for was addressed (lines 515-521), taking into consideration the response rates.
Reviewer 4 Report
Comments and Suggestions for Authors
The authors present a narrative review on surgical treatment of HCC. This is a somewhat well written manuscript which however repeats what has been published multiple times in the literature. It bring little new information.
Author Response
The authors present a narrative review on surgical treatment of HCC.
Point 1: This is a somewhat well written manuscript which however repeats what has been published multiple times in the literature. It bring little new information.
- This review was written to address surgical advancements in HCC treatment over the last decade; therefore, much of it is information previously published in the literature. In this review, we focused on recent advancements in the field such as ex vivo resections, associating liver partition and portal vein ligation for staged hepatectomy (ALPPS), and irreversible electroporation (IRE) to bring a sense of novelty to the article.
Round 2
Reviewer 4 Report
Comments and Suggestions for Authors
Once again, this review is weak in presenting novel information for the management of HCC. However, speaking of the comments by the authors who state that in this review, they focused on RECENT advancements in the field such as ALPPS for instance, in this case they have only cited 1 single original study (2021) and 1 case series with 2! cases (2019).